# Complementarity in Allen's and Bergmann's rules among birds

Justin W. Baldwin [1,5] ✉, Joan Garcia-Porta[1,2,3,5] & Carlos A. Botero [1,4]

Biologists have long noted that endotherms tend to have larger bodies (Bergmann's rule) and shorter appendages (Allen's rule) in colder environments. Nevertheless, many taxonomic groups appear not to conform to these 'rules', and general explanations for these frequent exceptions are currently lacking. Here we note that by combining complementary changes in body and extremity size, lineages could theoretically respond to thermal gradients with smaller changes in either trait than those predicted by either Bergmann's or Allen's rule alone. To test this idea, we leverage geographic, ecological, phylogenetic, and morphological data on 6,974 non-migratory terrestrial bird species, and show that stronger family-wide changes in bill size over thermal gradients are correlated with more muted changes in body size. Additionally, we show that most bird families exhibit weak but appropriately directed changes in both traits, supporting the notion of complementarity in Bergmann's and Allen's rules. Finally, we show that the few families that exhibit significant gradients in either bill or body size, tend to be more speciose, widely distributed, or ecologically constrained. Our findings validate Bergmann's and Allen's logic and remind us that body and bill size are simply convenient proxies for their true quantity of interest: the surface-to-volume ratio.

Geographic gradients in temperature have had profound impacts on the evolution of Life on Earth. For example, two of the most widely recognized rules in biology state that endotherms are likely to have larger bodies (Bergmann's rule[1]) and shorter appendages (Allen's rule[2]) in colder climates at higher latitudes or elevations (Fig. 1). Both Bergmann and Allen initially explained their observations by noting that an organism's surface-to-volume ratio (and therefore its capacity to lose heat) can increase, respectively, with smaller bodies and longer appendages. A wide variety of taxa conform to these expectations[3–5] but many conspicuously do not[6–10]. The causes for such discrepancies are not currently understood but are presumed to be related to methodological differences across studies[11], differences in sample sizes or geographic/thermal range coverage[12] and the idiosyncrasy of natural history traits[10].

Despite clear mechanistic and phenomenological similarities, Allen's and Bergmann's rules have often been evaluated independently[13,14]. Given that recent global warming has both reduced body size and lengthened appendages in a variety of organisms[15–17], we consider here that the evolutionary trajectories of body size and appendage length are likely to be related (see[18]). A potential complementarity between Bergmann's and Allen's rules suggests that pronounced latitudinal or elevational changes in appendage size could preclude the need for major variation in body size (or vice versa) (Figs. 1a, b and 2a, b). Alternatively, complementary changes in these

[1]Department of Biology, Washington University, St. Louis, MO 63130, USA. [2]Departament de Genètica Microbiologia i Estadística, Facultat de Biologia & Institut de Recerca de la Biodiversitat (IRBio), Universitat de Barcelona, Barcelona, Spain. [3]Department of Biodiversity, Ecology and Evolution, Complutense University of Madrid, Madrid, Spain. [4]Department of Integrative Biology, University of Texas at Austin, Austin, TX 78712, USA. [5]These authors contributed equally: Justin W. Baldwin, Joan Garcia-Porta. ✉e-mail: jwbaldwin@wustl.edu

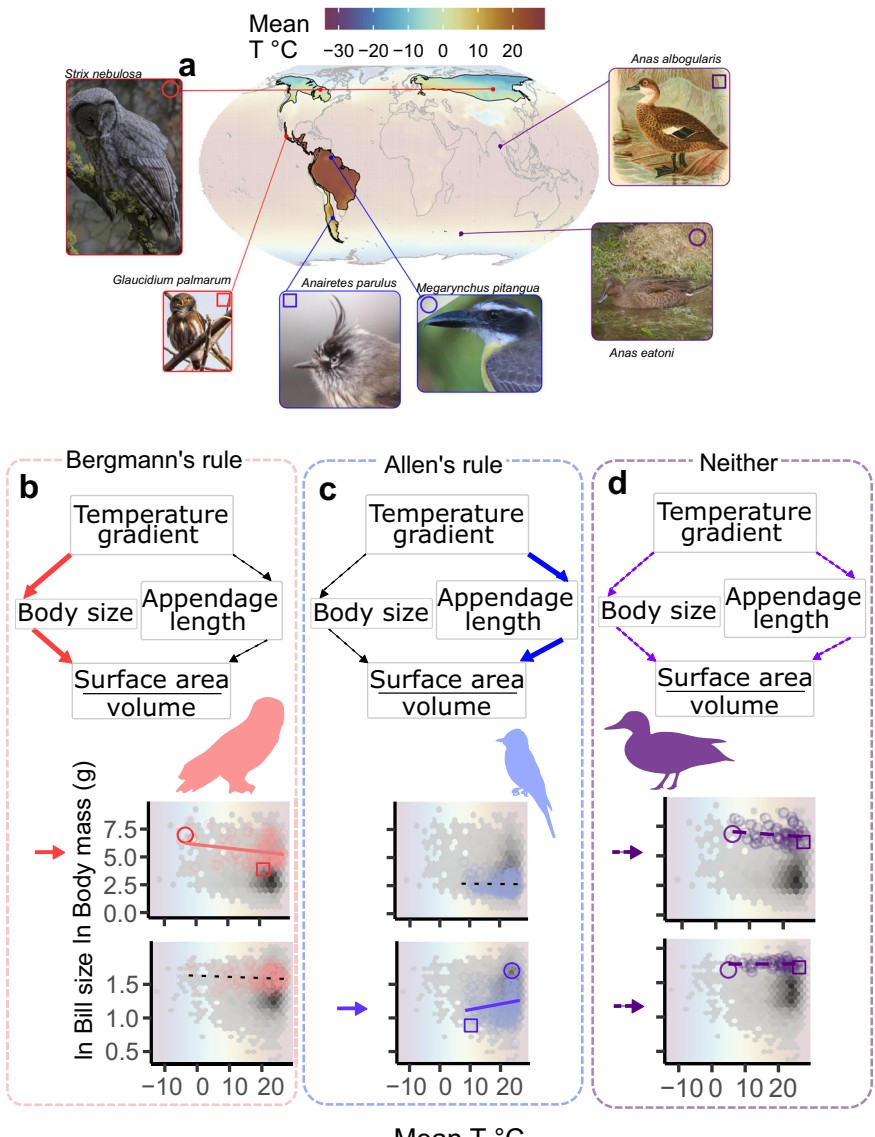

**Fig. 1 | Complementarity between Bergmann's and Allen's rules.** While some bird families conform to either Bergmann's rule or Allen's rule, most families conform to neither. For example, while owls exhibit significant changes in body but not bill size (red in **a** and **b**) and flycatchers exhibit significant changes in bill but not body size (blue in **a** and **c**), ducks exhibit instead complementary changes in both body and bill size that are subtle and difficult to detect statistically even though they exhibit the same trends that were predicted by Bergmann and Allen (purple in **a** and **d**). Symbols in scatterplots depict the species highlighted in **a**. Regression lines highlight conformance to rules (solid red/blue−significant conformance; loosely dashed black− non-significant change in one of bill or body size; densely dashed purple−non-significant trend in both bill and body sizes). We thank Gregory "Slobirdr" Smith, Ayna Cumplido, Félix Uribe, N. Hanuise, Ron Knight, John G. Keulemans, xgirouxb, and Andy Wilson for making their artwork and photos available on Wikimedia Commons and Phylopic under Creative Commons license CC-BY-SA (see Supplemental Information[52]; https://creativecommons.org/licenses/by-sa/4.0/).

traits could enable comparable changes in surface-to-volume ratio with more subtle phenotypic variation than predicted from either Bergmann's or Allen's rule alone (Fig. 1c, Fig. 2c). The latter strategy could presumably reduce the negative consequences of changing ecologically important traits[19]. For example, the significant body size reductions predicted by Bergmann could reduce inter-[20] and intra-specific competitive ability[21], whereas the changes in bill size predicted by Allen could alter foraging efficiency[22,25] and acoustic signaling[19,26–28].

To investigate the potential complementarity between Bergmann's and Allen's rules we present here a taxonomically comprehensive study of variation in morphology[29], climate niche[30], and geographic distribution[31] that includes 6974 species of terrestrial, non-migratory birds in 107 families (i.e., 89% of all terrestrial avian families with >10 species, i.e., 64% of all terrestrial birds[31,32]). We have chosen

this level analysis because family groupings typically offer a reasonably large number of species (which is needed for properly testing bio-geographic rules) and relatively low levels of variation in morphology, behavior, and ecology. Nevertheless, we note that the findings reported below remain unchanged when using higher taxonomic categories for our groupings (see Supplementary Fig. 5, Supplementary Table 2).

We begin by running two phylogenetic linear mixed models[33] to independently estimate interspecific variation in body and bill size within families, across geographic gradients of mean annual temperature. We then use the family-wide rates of change obtained in our initial analyses to explore the potential correlation between bill and body size changes. Finally, we investigate whether conformity to Bergmann's and Allen's rules is related to functional traits, extent of thermal ranges, and the number of species in a family. Our analyses

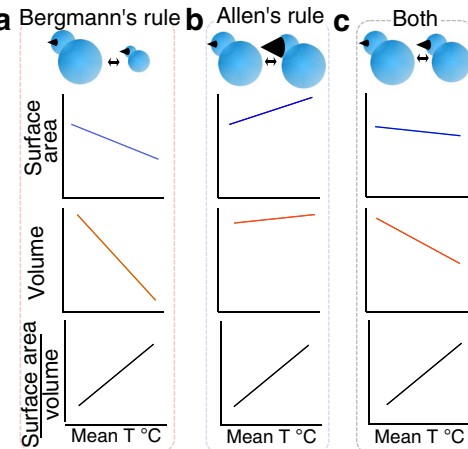

**Fig. 2 | Potential effects of complementarity in Bergmann's and Allen's rules.** Bergmann noted that changes in surface and volume typically accrue at different rates such that when body size decreases, the surface-to-volume ratio, SVR−and thereby the ability to dissipate heat−also increases (**a**). Similarly, Allen noted that appendages like the beak already have high SVRs so that when they become larger, SVR also increases (**b**). Here we note that by combining small changes in both traits, lineages can achieve comparable changes in their SVR without drastically altering their morphology and, presumably, their ecology (**c**). Depicted examples were simulated by approximating a bird's body with two spheres and one cone (cartoon depictions were drawn to exemplify the potential subtlety of these changes). Parameter values: Body size reduction factor in **a** = 1%; Beak size increase factor in **b** = 13.7% (volume to volume); Body size reduction factor in **c** = 0.5%; and beak size increase factor in **c** = 5.5%.

target the bill as the appendage of interest because of its well-known thermoregulatory function, high degree of vascularization[34], typical lack of thermo-insulation (i.e., no feather coverage), and proximity to the brain, one of the most energetically active tissues of the body[35]. Additionally, we note that in contrast to other extremities, the avian bill is hollow and lighter than other body parts of similar size, meaning that its potential contribution to thermoregulation from surface heat exchange is likely to be underestimated from overall body mass.

Overall, our study shows that geographic bill and body size gradients complement each other in terrestrial, non-migratory birds to enable smaller than expected morphological responses to temperature change. Additionally, it shows that either one of these typically weak geographic gradients is more likely to be detected as significant if statistical power is large, or if change in the other trait is constrained by ecology. These findings highlight the multivariate nature of selection, and remind us that body size and extremity size are simply convenient proxies for what both Bergmann and Allen envisioned as the likely target of selection from ambient temperatures: an individual's surface-to-volume ratio.

## Results and discussion
### Complementarity in avian bill and body size change
Because of the clear correlation between bill and body size (Fig. 3a), some earlier tests of Allen's rule in birds have relied on relative bill size as a metric of appendage length (i.e., the residuals from a regression of bill on body size). Accordingly, we estimated family-wide rates of variation in relative bill size across thermal gradients using a phylogenetic hierarchical regression model of bill size as a function of body mass, mean annual temperature, and diet, with phylogenetically pooled random intercepts and random slopes of temperature by family. Allometric scaling was investigated here as a non-linear effect (Δ AIC between linear and quadratic models[36] of bill size vs. body mass = 933, Supplementary Table 1, see[37]) and information on diet was

obtained from AVONET[29]. Through this model, we found that 81 out of 107 taxonomic families (i.e., 76%) do not exhibit significant conformance with either Bergmann's or Allen's rules (Fig. 3b, Table 1), where conformance is defined as a credible interval for a slope estimate that exhibits the expected direction and does not include zero. Intriguingly, the two families that exhibited conformance to both rules in this analysis did not show significant variation in absolute bill size over thermal gradients (Supplementary Fig. 2). Thus, the apparent increase in relative bill size observed in these families was driven not by Allen's expectation of enlarged bills, but rather by a relative reduction in body size (i.e., an indirect effect of Bergmann's rule[34]). This observation led us to re-evaluate the appropriateness of our proxy for estimating how appendages contribute to thermal adaptation.

Both body and bill size are easy-to-measure metrics that can be used to estimate what Bergmann and Allen intuited was the actual trait driving the patterns they observed (i.e., surface-to-volume ratio). However, because bird bills are hollow and weigh considerably less than other body parts of similar size, mass can grossly underestimate the bill's contribution to an individual's heat transfer capability. In that context, the missing component of interspecific differences in heat transfer capability may be more appropriately captured by absolute bill size, which in contrast to relative bill size, is proportional to surface area (Supplementary Figs. 3–4). After revisiting our initial analysis with absolute bill size, we discovered that the number of lineages that do not conform to Bergmann's and Allen's rules increased to 89 out 107 families (81%, Table 1) and that no single family showed simultaneous conformance to both rules. However, after accounting for uncertainty in the strength of both rules within a meta-regression framework using slopes from all families in the study, we found that more pronounced rates of change in body size are now tightly correlated with more muted rates of change in absolute bill size (Fig. 3c; slope posterior median = −0.233; 95% CI: −0.156−−0.320). Subsequent checks indicated that this finding is robust to the use of alternative data sources and methods for measuring bill size (Supplementary Fig. 5a−d, Supplementary Table 2), the inclusion/exclusion of families with narrow temperature ranges (Supplementary Fig. 5e, f), the threshold number of species per family used for inclusion in our analyses (Supplementary Fig. 5g, h), the use of higher taxonomic categories (Supplementary Fig. 5i, j), the consideration of phylogenetic uncertainty (Supplementary Fig. 6), and the inclusion of families with extreme bill morphologies (Supplementary Figs. 7, 8). Simply put, our analyses strongly suggest that most avian lineages have expanded their thermal ranges by evolving small and complementary changes in bill and body size rather than evolving pronounced changes in only one of these traits. Our findings are therefore consistent with both Allen's and Bergmann's observations and help explain why independent evaluations of these rules often fail to reach statistical significance (Fig. 3c).

### Apparent conformity to Allen's/Bergmann's rules
Having observed that most families do not exhibit pronounced morphological changes in bill or body size across thermal gradients, we now focus our attention on the few that do (Fig. 4, Supplementary Fig. 9). Specifically, we now investigate whether particular family characteristics determine the strength of morphological changes across thermal gradients, or the likelihood that we can detect such changes with traditional statistical methods. One possibility is that conformance to Allen's or Bergmann's rule is simply easier to detect when either statistical power or effect sizes are large[12]. For example, even weak gradients could be more easily distinguished from noise at larger sample sizes (i.e., in larger families). Similarly, interspecific differences could be more pronounced (and hence easier to detect) in families that cover wider temperature gradients or in families with bigger bodies or bigger beaks (because larger lineages need larger size changes to achieve a given percent difference in surface-to-volume ratio). Alternatively, major changes in one trait could be more likely to

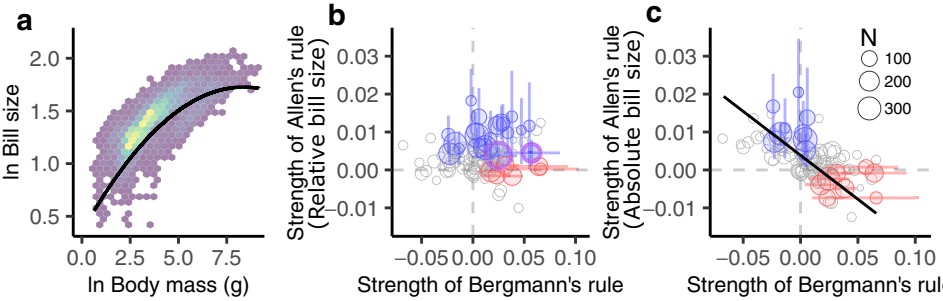

**Fig. 3 | Bill and body size variation in birds reveals alternative pathways of thermal adaptation. a** Bill size scales allometrically with body size (brighter colors indicate higher species counts; $N = 6,974$). **b** In analyses with relative bill sizes, we detect no correlation between the strength of Bergmann's rule (i.e., −1 times the family-wise slope estimates from a model of body size as a function of mean annual temperature) and that of Allen's rule (i.e., family-wise slope estimates from a model of bill size as a function of mean annual temperature; metaregression slope = 0.006; 95% CI = −0.074 to 0.089) and we find that 17 families conform to Allen's rule (blue circles), 2 conform to Bergmann's rule (red circles), 2 conform to both rules (purple), and 78 conform to neither (gray). **c** In contrast, similar analyses with absolute bill size, a better proxy for surface area, indicate that more pronounced changes in one trait are correlated with more muted changes in the other (solid black line depicts the metaregression fit: slope = −0.232; 95% CI = −0.322 to −0.150). In this version of our analysis, the number of families that do not exhibit any significant morphological changes increases to 87 and no families simultaneously conform to both rules. Circle sizes in **b** and **c** depict the number of species sampled within a family, whereas vertical and horizontal whiskers depict the 95% posterior credible interval of the estimated slopes.

**Table 1 | Family-level conformance to Bergmann's and Allen's rules across global temperature gradients**

| | Conformity to Allen's rule | | | | | |
|---|---|---|---|---|---|---|
| | Slope in relative bill size | | | Slope in absolute bill size | | |
| Slope in body size | Non significant | Positive (Allen's rule) | Negative | Non significant | Positive (Allen's rule) | Negative |
| None | 78 (40) | 17 | 1 | 88 (36) | 7 | 1 |
| Negative (Bergmann's rule) | 5 | 2 | 1 | 5 | 0 | 3 |
| Positive | 1 | 2 | 0 | 1 | 2 | 0 |

Cell counts show number of families that conform or not to the rules. Of the slopes found non-significant for both criteria, the count of families with slope values nevertheless in the expected direction for both biological rules is shown in parentheses.

be observed when changes in the other trait are difficult to achieve. For example, small-bodied lineages might already be near a lower limit for body size and may therefore need to adjust their surface-to-volume ratio through changes in bill size alone. Similarly, large-billed lineages may find it easier to evolve pronounced changes in body size because further enlargement of their beaks would require additional reinforcements of the skull, enlargement of jaw muscles, and other potentially costly features[37,38]. Given that bill morphology is critical to foraging success and tends to experience strong multivariate selection[19,28], it is also plausible that bill specialization constrains the evolution of bill size[23,24] and thereby favors a more pronounced evolution of changes in body size across thermal gradients.

To formally test these ideas, we investigated how temperature range, number of sampled species, mean body mass, mean absolute bill size, mean relative bill size, and mean degree of bill specialization influence not only a family's strength of conformity to Allen's and Bergmann's rules, but also the likelihood that these patterns will appear to be significant in statistical tests. We quantify bill specialization here as the rarity of a bill's shape, estimated through its 2D kernel density (see methods) within the morphospace defined by the first two principal components of the most comprehensive characterization of avian bill morphology to date[39] ($N = 3,512$ species, Fig. 4a–c, Supplementary Table 4). As expected, phylogenetic regression[36] indicates that families with larger bodies tend to exhibit more pronounced Bergmann's rule (Supplementary Table 5; Supplementary Fig. 10). However, opposite to expectation, a similar model revealed that the strength of Allen's rule decreases with temperature range, suggesting that the occupation of very wide temperature gradients relies less heavily on changes in bill size (at least exclusively so) than previously

thought. Additional randomization tests (see methods) indicate that families that do not exhibit significant conformity to either rule have smaller temperature ranges (one-sided tail in probability distribution beyond the observed value, $p = 0.001$) and fewer sampled species than expected by chance ($p < 0.001$). These analyses also indicate that families that exhibit significant conformity to Allen's rule tend to include a larger number of sampled species ($p < 0.001$), and to exhibit smaller bills ($p = 0.030$), and more common bill shapes ($p = 0.018$) than expected by chance. Similarly, families that exhibit significant conformity to Bergmann's rule tend to occupy larger temperature gradients ($p < 0.001$), to include more sampled species ($p < 0.001$), and to exhibit less common bill shapes ($p = 0.006$; Fig. 4, Table 2) than expected by chance. For example, we note that body changes over thermal gradients are most evident in bill specialists like hummingbirds, hawks, falcons, and owls. Taken together, these findings indicate that most often, a significant conformance to either Allen's or Bergmann's rules does not necessarily imply that a group is more strongly affected by temperature variation, but rather that it exhibits characteristics that facilitate the detection of what are typically very weak patterns. However, they also demonstrate that if the evolution of one trait is ecologically or phylogenetically constrained, then adaptation to temperature gradients is likely to lean more heavily on size changes in another trait.

In conclusion, we have shown that although geographic variation in bill and body size match theoretical expectations for adaptive thermoregulation in a comprehensive sample of birds, the complementary nature of these changes has enabled most lineages to locally finetune their potential for heat exchange with smaller-than-expected changes in either trait. These findings highlight the

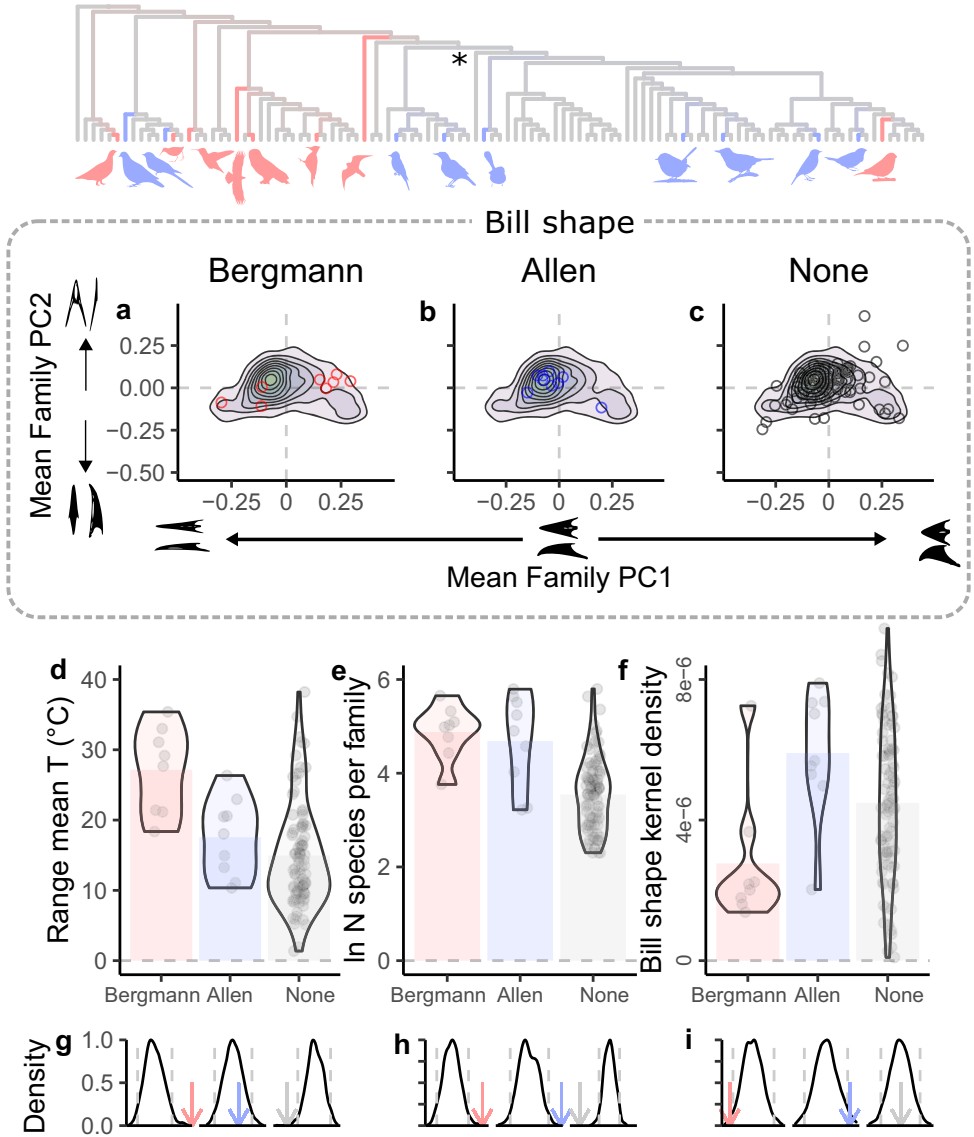

**Fig. 4 | Family-level traits are associated with conformity to Bergmann's and Allen's rules.** Cladogram without branch-length information showing family-level conformity to Bergmann's (red) and Allen's rule (blue). Icons depict representative taxa from conforming families and the asterisk denotes the ancestral node of passerines. Bill specialization can be estimated through bill shape characterization through a geometric morphometrics approach (see methods, i.e., contour lines for kernel densities in **a**–**c** indicate the rarity of a given shape). Silhouettes represent bill shapes that correspond to positions in PC 1 and 2. Families that conform to Bergmann's rule (red circles) tend to occur in the periphery of bill morphospace (**a**), whereas families that either conform to Allen's rule (blue circles) or neither rule (gray circles) tend to be more centrally located (**b**, **c**). Randomization models indicate that family-level factors that increase statistical power (i.e., extent of the temperature gradient for Bergmann, **d**, and sample size for both Bergmann and Allen, **e**) or that indicate bill specialization (**f**) tend to be higher in families that conform to these biological rules. Column heights in **d**–**f** depict family means and violin plots depict variation in the number of species per family across the range of each variable of interest. Randomization tests (**g**–**i**) confirm findings in **a**–**f** by showing that the corresponding observed values for these traits (arrows) lie outside the 95% interval of their expected null distribution (gray dashed lines). We thank Andy Wilson, xgirouxb, Ferran Sayol, Margot Michaud, Liftarn, Abraão B. Leite, Michael Scroggie, Metallura and Martin Bulla making their artwork available on Phylopic under Creative Commons licenses (CC-BY-SA, CC-BY, CC0−see Supplemental Information[52]; https://creativecommons.org/licenses/by-sa/4.0/).

multidimensional nature of adaptation, remind us that both body and bill size are simply convenient proxies for surface-to-volume ratio (i.e., the true quantity of interest behind Bergmann's and Allen's rules) and help explain why empirical validation of these patterns has frequently proven elusive (particularly when tested independently). In moving forward, we consider it imperative to formally recognize that Bergmann's and Allen's rules are different expressions of a common phenomenon (i.e., that size changes can alter a lineage's capacity for heat exchange, see[18]), and to be more careful in considering the limitations of our proxies until more

accurate estimates or more cost-effective measurements of surface-to-volume ratios become available.

## Methods
### Data and data sources
We obtained a time series of global climate (excluding Antarctica) from EcoClimate.org[30] covering the period from 1850 to 2005 at a spatial resolution of 0.5° × 0.5°. Using the Wagner IV equal area projection, we then extracted and calculated the mean annual temperature for the breeding range of each species in our data set from

**Table 2 | Family traits are associated with conformity to Bergmann's and Allen's rules**

| Trait | Conformance | Expected mean | Observed mean | P |
|---|---|---|---|---|
| Mean body mass | | | | |
| | Bergmann | 3.919 | 4.666 | 0.050* |
| | Allen | 3.896 | 3.256 | 0.059 |
| | Neither | 3.905 | 3.902 | 0.466 |
| Mean absolute bill size | | | | |
| | Bergmann | 1.428 | 1.501 | 0.161 |
| | Allen | 1.426 | 1.287 | 0.030* |
| | Neither | 1.426 | 1.433 | 0.212 |
| Mean relative bill size | | | | |
| | Bergmann | 0.123 | 0.106 | 0.353 |
| | Allen | 0.123 | 0.088 | 0.194 |
| | Neither | 0.123 | 0.128 | 0.169 |
| Temperature range | | | | |
| | Bergmann | 16.182 | 27.153 | <0.001* |
| | Allen | 16.150 | 17.557 | 0.284 |
| | Neither | 16.082 | 14.966 | 0.001* |
| ln N species per family | | | | |
| | Bergmann | 3.755 | 4.880 | <0.001* |
| | Allen | 3.732 | 4.685 | <0.001* |
| | Neither | 3.740 | 3.544 | <0.001* |
| Mean kernel density | | | | |
| | Bergmann | 4.43E-06 | 2.77E-06 | 0.006* |
| | Allen | 4.50E-06 | 5.68E-06 | 0.018* |
| | Neither | 4.48E-06 | 4.50E-06 | 0.464 |

Custom randomization tests (see methods) were performed by 1000 iterations of uncoupling family traits and conformance categories and calculating the expected distribution of family-level trait values and comparing them to observed group-level means ($N = 107$). P-values indicate the one-sided proportion of expected distributions beyond the observed means (Fig. 3g–i). Significant differences between observed and expected means are indicated with asterisks.

distribution limits downloaded from BirdLife International[31] on 13 Feb 2019. Phylogenetic hypotheses for our analyses were downloaded from the 2016 version of the Global Phylogeny of Birds[32] from which we generated a summary tree following previous work[40]. Migratory species, conservatively defined here as the 1,879 species that occupy breeding localities for only a fraction of the year, were excluded from our analyses because the range of temperatures they experience cannot be accurately measured solely from their breeding range. Similarly, our analyses exclude families with <10 year-round resident species sampled or available.

Following earlier global analyses on birds[4], we quantified body size as the mean log-transformed average body mass for both sexes (data from[41]). We favored a mass-based metric of body size (as opposed to the osteological estimates that are typically used in smaller scaled studies[42,43]), because many of the non-Passerine orders in our sample are subject to constraints on tarsus length (e.g., exclusively fossorial ratites and primarily aerial Apodiformes[44,45], Supplementary Fig. 1). Overall, the mean body mass in our sample ranged from 1.9 g (Trochilidae: *Thaumastura cora*) to 9798 g (Accipitridae: *Gyps himalayensis*).

We used two alternative methods to quantify the high-dimensional nature of bill size in our analyses. The first metric was derived from four linear measurements of bill size collated from AVONET[29]: length of exposed culmen (EC), tip-to-nares distance (TND), bill width, (BW) and bill depth (BD) ($N = 6,974$ species). Because linear dimensions of the bill tend to be highly correlated, we first transformed them using the Box-Cox transformation[46] and them reduced

them to a single composite metric using a correlation-based principal component analysis (PCA) in the R package psych[47]. Our first estimate of bill size is therefore the first unrotated principal component from this PCA, which captures 79.93% of the total variation in linear bill measurements (PC1 loadings: EC = 0.89; TND = 0.90; BW = 0.89; BD = 0.90). Our second estimate of bill size was independently derived from a dataset[39] of tridimensional landmark configurations (79 landmarks) that describe beak shape for 5,551 species, of which 3,512 were retained in the main analysis. In this case, we computed and subsequently used the centroid size[48] for each landmark configuration using the R package geomorph[49].

**Quantifying rates of morphological change**

We estimated Bayesian phylogenetic linear mixed models, hereafter BPLMM, in the R package MCMCglmm[33] to determine whether the mean annual temperature of a species' breeding range is related to the log-transformed values of either its body size or bill size. Support for Allen's and Bergmann's rules was assessed by evaluating whether the 95% credible intervals of family-level random slopes ($\beta_{\text{Temperature},j}$) did not include zero and matched expectation (i.e., positive slopes for Allen's rule and negative for Bergmann's). Given that diet is known to influence bill size[23], each model of (relative and absolute) bill size included a 5-level diet category as an additional predictor variable. Thus, the BPLMM for Bergmann's rule was defined as,

$$\ln(\text{Body Mass}_{i,j}) = \alpha_j + \beta_{\text{Temperature},j} \times \text{Temperature}_{i,j} + \varepsilon_i \quad (1)$$

where $j$ denotes family and $i$ denotes species. Herein, the family-level intercepts $\alpha$ follow $\alpha_j \sim N(\mu_\alpha, \sigma_\alpha^2 \times \Omega)$, and the family-level slope of body size change over temperature $\beta_{\text{Temperature}}$ follow $\beta_j \sim N(\mu_\beta, \sigma_\beta^2 \times \Omega)$, where $\Omega$ is a variance-covariance matrix derived from tip-distances of an ultrametric phylogenetic family tree[32,50]. The residuals are $\varepsilon_i \sim N(0, \sigma_o)$ where $\sigma_o$ represents the residual sampling variance. Moreover, $\sigma_\alpha^2$ and $\sigma_\beta^2$ are the phylogenetic variance of the intercepts and slopes, respectively. Intercepts are slopes were set to follow an unstructured correlation structure, where $[\mu_\alpha, \mu_\beta] \sim \text{MVN}\left(0, \begin{matrix} \sigma_{\mu\alpha} & \sigma_{\mu\alpha\beta} \\ \sigma_{\mu\alpha\beta} & \sigma_{\mu\beta} \end{matrix}\right)$. The BPLMM for Allen's rule was similar, except for an added term for diet-based variation, as follows:

$$\ln(\text{Bill size}_{i,j}) = \alpha_j + \beta_{\text{Temperature},j} \times \text{Temperature}_{i,j} + \text{Diet}_{i,j} + \varepsilon_i \quad (2)$$

To determine whether the allometry between bill and body size was nonlinear, we performed two phylogenetic linear regressions using phylolm and a species tree from[32,50]. The nonlinear allometry explained 14.4% more variation than the linear model (Supplementary Table 1). BPLMMs were also used to account for the nonlinear allometry of avian bills and bodies (Fig. 3a). In this version of our analyses, our model of log bill size simply included mean breeding range temperature, as well as log body size and a quadratic effect of log body size as predictors (all other details of the model remained unchanged):

$$\ln(Billsize_{i,j}) = \alpha_j + \beta_{\text{Temperature},j} \times \text{Temperature}_{i,j} + \beta_{\text{BodyMass}} \times \text{BodyMass}_{i,j} + \beta_{\text{BodyMassSquared}} \times \text{BodyMassSquared}_{i,j} + \text{Diet}_{i,j} + \varepsilon_i \quad (3)$$

We fitted three independent chains for every BPLMM and assessed their convergence through Gelman-Rubin statistics <1.1, effective sample sizes near 1,000 for every parameter, and visual inspection of trace plots. Each model chain was run for 13,000 iterations with a "burn-in" of 3,000, a thinning interval of 10, and uninformative priors that included inverse Wishart distributions with $\psi = 1$ and $\nu = 0.02$ for the random effects. Variance inflation factors were also computed for every model with more than one predictor and were determined to be

<1.1, which is well below the often used cutoff of ten[51] (R code on Zenodo[52]).

## Relative or absolute bill size?

Relying on relative bill sizes when testing Allen's rule seems very intuitive, but there are at least three potential concerns with such practice. Technical arguments have been made elsewhere[53,54] so we focus here instead on how apparent conformance to Allen's rule can sometimes occur in the absence of bill size enlargement if relative bill sizes are used and on how relative bill sizes are not always correlated with surface area (i.e., can lead to erroneous inferences on the bill's actual contribution to heat dissipation).

**Bill size increases or body size decreases?.** Other scholars have noted that relative bill sizes can lead to incorrect perceptions of conformance to Allen's rule if appendage sizes remain unchanged but body sizes decrease over thermal gradients[34]. Among birds, this scenario could be expected in taxa with extreme bill morphologies given that their bill evolution is presumably constrained by their atypical ecologies (see main text). Our data support this view. Specifically, hummingbirds and rails exhibit uncommon bill morphologies and strong conservation of bill size across thermal gradients (Supplementary Fig. 2). Because these two clades conform to Bergmann's rule, an analysis of their relative bill size gradients suggests that the length of their extremities increases in warmer climates. These findings confirm that alternative metrics of bill size can emphasize very different aspects of a bill's contribution to heat exchange, highlighting the importance of choosing a metric that more closely estimates the independent contributions of body and bill to surface-to-volume ratio when testing the interaction between Allen's and Bergmann's rules.

**Proxies of surface area.** Surface area is typically correlated with absolute bill size but not necessarily with relative bill size (Supplementary Fig. 3). The data presented here were obtained through simulation assuming simplified conical bill shapes with base radii that are proportional to cone height (R code on Zenodo[52]). Empirical data also support the notion that absolute bill size is a better proxy of the capacity to transfer heat than relative bill size. For example, the Toco Toucan (Rhamphastidae: *Ramphastos toco*) and the Purple-crowned Fairy (Trochilidae: *Heliothryx barroti*) have almost identical relative bill sizes (residual bill size = 0.36) but dramatically different bill surface areas and heat dissipating capabilities (Supplementary Fig. 4). For example, the Toco Toucan loses up to 400% of its resting heat production through its bill[55], whilst the Calliope Hummingbird (*Selasphorus calliope*–a migratory hummingbird that was not included in our analysis but shares many family-wide conserved characteristics with the Purple-crowned Fairy) chiefly dissipates heat through axial areas, feet and eyes[56].

## Robustness analyses

In the following subsections we explore the robustness of our findings to a series of methodological choices and sources of uncertainty.

**Phylogenetic principal components analysis.** We use PCA in the main text strictly to reduce the dimensionality of alternative linear metrics of bill size. Nevertheless, given that the relationships between different dimensions of a bird's beak can be phylogenetically constrained, we consider here the possibility that our findings may be affected by the phylogenetic non-independence of the variables included in the PCA[57]. The PC1 scores derived from a phylogenetic PCA with Brownian model of evolution and a non-phylogenetic PCA are highly correlated (Pearson's product moment correlation: t-statistic (df = 6972) = 2575.2, $p < 0.001$, $\rho = 0.99947$, 95% confidence intervals = 0.99945 to 0.99950, Supplementary Fig. 5a). Most importantly, our findings are qualitatively identical when downstream analyses are

performed with either metric (compare Fig. 3c with Supplementary Fig. 5b; Supplementary Table 2).

**Estimates of bill size.** Size estimates derived from linear metrics can sometimes be very different from those derived from geometric morphometric analyses. In this case, linear estimates of bill size were highly correlated with centroid-distance-based estimates (Supplementary Fig. 5c). Here too, our main findings are qualitatively identical for both metrics (Supplementary Fig. 5d; Supplementary Table 2).

**Thermal range cutoffs.** Geographically restricted bird families can often exhibit low variance in climate and may therefore be less likely to exhibit the patterns that Bergmann and Allen predicted. We addressed this issue by repeating our analyses with the subsample of families that individually span a range of mean temperatures of at least 10 °C (Supplementary Fig. 5e). Our general findings remain unchanged (Supplementary Fig. 5f; Supplementary Table 2).

**Species sampling cutoffs.** Because poorly sampled families could theoretically bias our analyses towards not detecting Allen's and/or Bergmann's rule, we repeated our analyses using a cutoff of 20 species instead of 10 per family (Supplementary Fig. 5g). Our general findings remain unchanged (Supplementary Fig. 5h; Supplementary Table 2).

**Taxonomical groupings.** To further investigate Allen's and Bergmann's rules in more speciose groupings, we also reran our analyses using biological Orders instead of Families. These alternative groupings yielded highly uneven sampling (Supplementary Fig. 5i), but our general findings remain unchanged (Supplementary Fig. 5j; Supplementary Table 2).

**Phylogenetic uncertainty.** Current phylogenetic hypotheses disagree on the placement of certain taxonomic groups (Supplementary Fig. 6a). We assessed the potential effect that these disagreements could have on our findings by repeating every phylogenetically informed analysis with alternative maximum clade credibility trees estimated from 1,000 randomly selected tree topologies with either the Hackett[50] or the Ericson[58] backbone. Our findings are not contingent on the phylogenetic hypothesis being used (compare Fig. 3c with Supplementary Fig. 6b). Furthermore, we find no significant differences between these alternative analyses in the proportion of families that exhibit significant conformity to Bergmann's and Allen's rules (Supplementary Table 2).

**Atypical morphology.** Our main analysis excluded the swifts (Apodidae) because of their atypical morphology (i.e., small beaks–Supplementary Fig. 7–and minimal tarsi–Supplementary Fig. 3) and their propensity to spend most of their time flying[44] at elevations where temperature can differ considerably from those at ground level (i.e., the level at which we temperature is sampled in the EcoClimate data set[30]). Nevertheless, including swifts in our analyses does not only lead to qualitatively identical findings (e.g., Supplementary Fig. 8), but also to further support for the notion that Bergmann's rule is more evident when the evolution of beaks is strongly constrained. Specifically, as expected from their highly specialized and reduced beak morphology[44] swifts do not conform to Allen's rule but exhibit instead pronounced variation in body size over thermal gradients (i.e., Bergmann's rule).

## Correlating changes of bill and body size

We used a regression method commonly used in meta-analyses to account for independent uncertainty in our observations of the predictor and response variables. First, we extracted for each family, the credible intervals for the random slope estimates of the effects of temperature on body and bill size from the hierarchical models

shown above in our methods. Subsequently, we evaluated the relationships between bill and body size gradients accounting for uncertainty in these estimates using a metaregression model[59] in brms[60,61] following established guidelines[62,63] (R code on Zenodo[52]). The meta-regression model was characterized by the following equations, where $\beta^{\text{bill}}_{\text{Temperature},j}$ are the random slope posterior point estimates of temperature on bill size of family $j$ extracted from the BPLMM of bill size and $\sigma^{\text{bill}}_{\text{Temperature},j}$ are the standard errors of its posteriors. Similarly, $\beta^{\text{body}}_{\text{Temperature},j}$ are the random slope posterior point estimates of temperature on body size of family $j$ extracted from the BPLMM of body size and $\sigma^{\text{body}}_{\text{Temperature},j}$ are the standard errors of its posteriors.

$$\beta^{\text{bill}}_{\text{Temperature},j} \sim N(\theta_j, \sigma^{\text{bill}}_{\text{Temperature},j})$$
$$\theta_j = \alpha + \beta_{\text{metaregression}} \times \beta^{\text{body}}_{\text{Temperature},j} \quad (4)$$
$$\beta^{\text{body}}_{\text{Temperature},j} \sim N(\mu_{\text{body}}, \sigma^{\text{body}}_{\text{Temperature},j})$$

We evaluated the correlation of strengths of Bergmann's and Allen's rule based on the posterior distribution of the metaregression parameter $\beta_{metaregression}$ and checked if its 95% credible overlapped with zero.

### Predictors of conformance to both rules

**Randomization tests.** We first visualized the phylogenetic distribution of slope estimates across the family tree (Supplementary Fig. 9; prepared with the R package ggtree[64]). Then we investigated the potential drivers of apparent conformance to Bergmann's and Allen's rules through a series of randomization tests on family-level traits. Mean relative bill sizes were computed by averaging the residuals for each family from a phylogenetic regression model[36] in which bill size was predicted by both a linear and a quadratic term for log body mass. Sample size was measured as the number of species sampled from each family, and temperature range was computed as the maximum mean breeding temperature observed in a family minus its minimum mean breeding range temperature. Bill specialization was quantified by computing and plotting the first two principal components of the geometric morphometric data obtained from[39], which jointly captured 86.59% of the total variation in bill size and shape (Supplementary Table 4). Subsequently, we used the resulting PC scores to estimate a two-dimensional kernel density function (Fig. 4a–c) that allowed us to distinguish rare bill shapes (i.e., low kernel densities) from more common ones (i.e., high kernel densities). Bill specialization was subsequently quantified as the family-level average of kernel densities in the sampled species from a family. We implemented a univariate approach to investigate potential relationships between family-level characteristics and conformance to Allen's and Bergmann's rules. We began these analyses by estimating, for each trait, the expected null distribution of family means in each conformance category through 1,000 randomizations of trait values among categories. We then assessed significant deviations from chance expectations by computing the cumulative probability density of mean values less than or equal to the observed mean. Observed means with cumulative probability densities below 2.5% or above 97.5% were interpreted as significantly different from random expectations. The p-values reported in Table 2 indicate probability densities in the area contained between the observed mean and the nearest extreme.

### Regressions of conformance intensity.

To evaluate the extent to which family characteristics influence the observed strength of Allen's and Bergmann's rules, we fitted phylogenetic regression models in which the slopes of either bill or body size changes over temperature gradients were evaluated as a function of mean absolute body size, mean relative bill size, temperature range, ln number of species per family, and the mean rarity of bill shape. Given the strong correlation

between body and bill size (see Fig. 3a), it was unfeasible to include both terms in a single model so we included mean bill size instead in the model of Allen's rule and mean body size in the model of Bergmann's rule (Supplementary Table 5; Supplementary Fig. 10). All resulting variance inflation factors were below 2 indicating no major concerns of multicollinearity[51]. We fitted both regression models using a family tree with the Hackett backbone[32,50] in phylolm[36] using Pagel's branch length transformation.

### Reporting summary

Further information on research design is available in the Nature Portfolio Reporting Summary linked to this article.

## Data availability

No new data were generated for this study. Avian morphology data is available from linear measurements[29] and geometric morphometric landmarks[39]. Temperature data are available from EcoClimate[30]. Distribution and ranges are available from BirdLife International[31]. Phylogenetic data are available from the Global Bird Phylogeny[32]. The analytic data are available on Zenodo (https://zenodo.org/record/8092265)[52].

## Code availability

The code is available on Zenodo (https://zenodo.org/record/8092265)[52].

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

## Acknowledgements

Thanks to Jonathan Losos for comments on an early draft. C.A.B. was funded by NSF, award number DEB 184147. J.G.-P. was supported by a Beatriu de Pinós fellowship from the Generalitat de Catalunya (2020 BP 00147) and by the program "Atracción de Talento Investigador Modalidad I" from the Comunidad de Madrid (2022-T1/AMB-24171).

## Author contributions

Conceptualization: J.W.B., J.G.-P., C.A.B. Methodology: J.W.B., J.G.-P., C.A.B. Investigation: J.W.B., J.G.-P. Visualization: J.W.B. Funding acquisition: C.A.B. Project administration: J.W.B., J.G.-P., C.A.B. Supervision: C.A.B. Writing—original draft: J.W.B. and C.A.B. Writing—review and editing: J.W.B., J.G.-P., C.A.B.

## Competing interests

The authors declare no competing interests.
