## [Peer Review File · Nature Communications]

Complementarity in Allen's and Bergmann's rules among birdsReviewers' Comments:

Reviewer #1:

Remarks to the Author:

This is a lovely paper by Baldwin et al. examining – in truly impressive detail – the application of Bergmann's and Allen's rules across a nearly comprehensive sample of non-migratory birds. They conduct their analyses at the family level, and show that Bergmann's and Allen's rules do generally apply at this scale, but very weakly, and often complementarily. They then correlate the strength of the Bergmann's and Allen's gradients with various ecological and evolutionary factors, finding that adherence to one or the other biogeographic rule is more likely in clades with more species and/or wider temperature niches, with Allen's rule more likely in clades with more common bill shapes and Bergmann's rule more likely in clades with less common bill shapes. They also demonstrate that it is the surface-to-volume ratio, rather than relative bill size per se, that is the variable that should be considered when testing Allen's rule, which is an elegant bit of insight.

I wish to especially praise the sheer amount of work that went into justifying these analyses; the authors have included a nearly-alarming volume of sensitivity analyses. I particularly like the use of both AVONET linear measurements and the Cooney et al. morphometric measurements, as well as the explicit consideration of where various bill shapes fall within morphospace. I also appreciate that the authors often restate their technical results in terms of the biological concepts of the system; this makes for very readable text, and I wish more in our field would do this.

I have two major comments, both of which are fairly philosophical, and then a small handful of minor comments.

The first major comment is that this analysis is conducted at the family level, for families with 10 or more species. I am worried that this may be insufficient statistical power to detect relevant trends, particularly in a phylogenetic framework. (I say this hypocritically, as I too have published on family-level analyses.) I would strongly encourage the authors to more explicitly explain why they think that "family" is the correct level to test their ideas at. I might also encourage (if it is not too much work), a set of analyses using a higher family size cut-off (20 species? even higher?), and/or to investigate what these patterns look like at the order level.

The second major concern is that a few of the authors' biological conclusions are not very clearly justified, potentially due to a word limit in the main text. Speculation on the many non-environmental pressures constraining beak size evolution, for example, or on why some clades adapt to thermoregulatory demands via Bergmann's Rule versus Allen's rule, is fascinating, and I think a real strength of this paper. More time explaining these assertions, however, would be prudent (especially lines 151-155).

Minor comments:

By omitting the migratory species, there is going to be substantial geographic bias – do the authors think that their tilting focus towards tropical species, and away from temperate species, is affecting their results in any way? (To be clear, I find the decision to omit the migratory species to be well-justified, I'm just wondering what the biological repercussions of this decision are.)

The figure quality in the reviewer copy is very poor, presumably something on the manuscript software's end; something to be aware of for future submissions!

L119 – 25 is a strange citation for the multivariate selection occurring on beaks. Did you perhaps mean <https://doi.org/10.1098/rspb.2019.2474>, which has some overlap in authors? (Alternatively, if

you want a citation on the competing effects of song and the environment on bird bill shape evolution, I quite like <https://doi.org/10.1111/ele.14102>) (And indeed, I don't think you mean citation 25 in line 51 either?)

L307-301 – Using whose classification of migratory species? (Extracted from BirdLife's ranges?)

L326: Indicate somewhere what EC, TND, BW, and BD mean (e.g., by putting the abbreviations in parentheses in lines 320-321).

Data availability: Though no new data in the strictest sense were generated for this study, a substantial amount of work went in to generating morphological PC scores and new environmental variables. I don't know the exact Nature Communications policy on data deposition, but I would expect these to be included for basic reproducibility and transparency.

Code availability: Including the code as a PDF is appreciated (it's more than many people do!), but it would be even better to deposit it in a format that can be easily copied and pasted. Furthermore, without the accompanying data (see above), it is not as helpful.

Code supplement: The use of red and green in the figures in the code supplement will make this very hard for readers with red-green colour-blindness to interpret; I strongly suggest selecting different colours.

Extended data figure 1: Are there photo credits for panel A? Or credits for the silhouettes in panels B-D?

Extended data figure 2: What does the vertical blue line for the Rheidae mean?

Reviewer #2:

Remarks to the Author:

NCOMMS-22-42817

Complementarity in Allen's and Bergmann's rules among birds

Broad comments:

In this paper, the authors undertake an interesting exploration of the combined influences of overall body size as well as the shape of organisms for thermal regulation. They seek to simultaneously estimate the support for two long-standing proposed macroecological gradients using existing morphological data from a large fraction of bird species from across the world. I appreciate the large-scale nature of this study and the efforts made to approach these questions from different angles using different morphological metrics, and their attempts to explain differences in responses among taxonomic groups. While I think there are interesting findings presented here, some quite crucial additional considerations are needed in multiple areas.

In particular, I had several major concerns pertaining to the bill length metrics used here and the implications these have for how results of this study are to be interpreted. Firstly, I have two points about relative bill size. There is a clear non-linear relationship between log bill size and log body mass (Fig. 2a). Modeling this using a linear regression therefore is not giving an accurate representation of relative bill size. Additionally, the authors make the case for the importance of bill surface area, so why not use that directly (estimated directly from bill measurements available in AVONET) rather than use length?

The authors ultimately conclude that relative bill length isn't the best metric to be used here, since any change in relative bill size might be due simply to changes in body size. I found it somewhat strange that such a large portion of the paper is dedicated to talking about relative bill size (throughout the manuscript) when this is ultimately rejected as being a valid approach by the authors

(their thoughts, though not necessarily mine). The authors decide to use absolute bill length as an alternative metric. However, there are other issues with the use of this metric. Absolute bill size is likely to vary with body size for reasons unrelated to temperature (e.g., developmental or dietary reasons). The fact that there is a negative relationship between bill size and temperature (i.e., opposing Allen's Rule) for bird families that show a strong negative relationship between temperature and body size (i.e., supporting Bergmann's Rule) supports this notion. That is, for families where species are larger in colder environments, they also tend to have larger bills in these environments (negative relationship between these two slope estimates as presented by L 93 and Fig. 2c). As traits tend to scale with body size, this isn't surprising. The question is, do these bill sizes vary more than would be expected under some null expectation of how bill size varies with body size? Without assessing deviations from this null expectation (what residuals are typically trying to get at), I don't think an accurate assessment of Allen's Rule can be made here. Given that this 'trade-off' between Bergmann's and Allen's Rules are a main focus of this paper, I think this stands as a major issue in this manuscript in its current form. I'm not sure there is a panacea to the fact that changes in relative bill size could be due to changes in body size alone. It does appear that species that have positive support for Allen's Rule (larger bills in warmer areas) tend to have positive support for Bergmann's Rule (smaller bodies in warmer areas) – the caption states that no correlation was found, but I was unclear what analysis was used to conduct this analysis (see also detailed comments below). Body size may simply be less evolutionarily constrained than bill size, due to things like dietary factors (bill size and shape have been shown to be important for foraging, e.g., Galápagos finches, hummingbirds) and bird song (due to the importance of sexual selection). I think something that could be presented is how relative bill size change is related to absolute bill size change to assess this (that is, a scatterplot with these two metrics on the two axes).

Additionally, the authors assume here that species are less impacted by shifting their morphology according to Allen's and Bergmann's Rules simultaneously (L 48, 106), compared to just one of these traits. I agree that changes in these traits could have important ecological implications (for several reasons). But if an organism is changing in two traits, then it is being impacted in two different ways. Just not as prominently in one single way (if there is some trade-off there, as we might expect). Is there really any buffering going on here by changing in two ways rather than one? That is, is it better to be slightly hampered in two different ways than more substantially hampered in one way? Possibly it is, but I don't think any evidence of this notion is presented here (nor can I think of a good way to actually explore the degree to which this might be true). I think changes in both size and appendage size are probably due to the fact that evolution acts on multiple axes at once. Maybe some species are more constrained in one area compared to another (possibly due to constraints due to diets, etc. as acknowledged in several places in the manuscript, including L 118). I would suggest rethinking how this is presented.

I do appreciate the consideration of how species' responses vary according to different traits and family characteristics. The idea that unusual bill sizes and shapes might be related to these patterns is interesting. The fact that families in warmer areas tend to correspond more with Bergmann's Rule is interesting. This suggests that it's these very warm areas that matter most for body size. This is in line with some recent work done within-species, where larger effects of temperature were found in warmer parts of species' ranges (Youngflesh et al. 2022) and on the consequences of warming for body size in desert environments (Riddell et al. 2019). I do, however, think the analyses presented in Fig. 2d/e/f/g/h/u should be analyzed using continuous values. That is, the actual slope estimate (with associated uncertainty) for the Rule support used as the response variable, while temperature/# of species/bill shape used as the response variable. While presented in terms of a binary classification, support for these rules is actually a continuous measure.

I think more clarity regarding the statistical models is needed as well (both the overall relationship between temperature and both body size and bill size as well as in assessing the relationship between support for Bergmann's Rule and Allen's Rule). That is the authors could explicitly write out the full statistical model used in mathematical notation. For example, $y_i \sim N(\alpha + \beta x_i, \sigma)$ would be used to denote a simple linear model, where i is the index for observation. I bring this up because the structure of the model has interpretations for how results are interpreted. I assume that a separate slope and intercept are estimated for each family in the initial analyses, but it's unclear in the text.

Overall, while I enjoyed aspects of this paper and appreciate many of the ideas presented, I think additional considerations are needed. In particular, in my opinion the authors should rethink how these questions are approached and how results might be interpreted because of this. I think additional polish of these ideas and of the presentation are necessary before this could be acceptable for publication.

Minor comments:

General: I noticed the citations numbers are off in the manuscript. For example, citation 36 is not about brain size at L 60 – I believe that should be citation 35.

General: I appreciate that the authors published the environment (session info with OS and package versions) used in the analyses. Information on the specific environment is not included enough, in my opinion. As a very minor point, I don't think all trace plots or Rhat values from the Bayesian model are needed in the supplement. I think including them distracts from the rest of the code.

Extended Data 1: Why have dashed lines only for ducks? Shouldn't there also be dashed lines for bill size in owls and body size for flycatchers?

Fig. 1: It's a bit difficult to tell that the size of the 'birds' is different as well in panel C. I might suggest making the difference more pronounced.

L 53: suggest removing 'comprehensive'. This is a large-scale study with a lot of value, but it is not a comprehensive analysis of morphology.

L 56: I would suggest specifically stating that you looked at variation across species, within families, rather than simply saying 'family-wide' to make it more clear what was done.

ED Fig. 4: I'm not sure the information presented in panel d is very useful. We wouldn't expect relative bill size to correspond to bill surface area, but to relative bill surface area. That might be a better metric to have on the y-axis.

ED Fig. 5: If this is using simulated data, this should be stated in the caption. Is there an assumption being made in the generation of these data that bill size varies with body size linearly? Is that a good assumption? Would we expect that? I'm not sure I would as bill length is a one-dimensional measure and mass is often considered to be proportional to volume, which is a three-dimensional measure (and these things tend to scale with their dimensionality).

L 426: I think it's important to write out this model explicitly, using mathematical notation. I assume that the authors model 'observation error' in both the response and predictor here (with the response being the posterior mean values and the 'error' represented as the posterior standard deviation [or variance if that is used for the subsequent model]), but this is somewhat unclear. Were all data points used here or just those deemed to 'have an effect'?

ED Table 1: What do these numbers represent? The number of families?

General: It might be nice to have a table in the supplement that describes how many species each family had in the filtered dataset, and perhaps the general characteristics of that family (thermal range, total area occupied by that family, etc.)

Fig. 3: I think the bill characteristic images on the x-axis are somewhat confusing, in that they are distributed across the three panels. Additionally, I'm not sure it's appropriate to display each species as a point here. Conformity to these Rules are characterized at the family, not species level. This is a

sort of visual pseudo-replication issue, in my opinion. Best to take the family average for these bill traits (which might also make the plots more clear).

References:

Riddell, E. A., K. J. Iknayan, B. O. Wolf, B. Sinervo, and S. R. Beissinger. 2019. Cooling requirements fueled the collapse of a desert bird community from climate change. *Proceedings of the National Academy of Sciences* 116:21609–21615.

Youngflesh, C., J. F. Saracco, R. B. Siegel, and M. W. Tingley. 2022. Abiotic conditions shape spatial and temporal morphological variation in North American birds. *Nature Ecology & Evolution* 6:1860–1870.

REVIEWER COMMENTS

Reviewer #1 (Remarks to the Author):

This is a lovely paper by Baldwin et al. examining – in truly impressive detail – the application of Bergmann’s and Allen’s rules across a nearly comprehensive sample of non-migratory birds. They conduct their analyses at the family level, and show that Bergmann’s and Allen’s rules do generally apply at this scale, but very weakly, and often complementarily. They then correlate the strength of the Bergmann’s and Allen’s gradients with various ecological and evolutionary factors, finding that adherence to one or the other biogeographic rule is more likely in clades with more species and/or wider temperature niches, with Allen’s rule more likely in clades with more common bill shapes and Bergmann’s rule more likely in clades with less common bill shapes. They also demonstrate that it is the surface-to-volume ratio, rather than relative bill size per se, that is the variable that should be considered when testing Allen’s rule, which is an elegant bit of insight.

I wish to especially praise the sheer amount of work that went into justifying these analyses; the authors have included a nearly-alarming volume of sensitivity analyses. I particularly like the use of both AVONET linear measurements and the Cooney et al. morphometric measurements, as well as the explicit consideration of where various bill shapes fall within morphospace. I also appreciate that the authors often restate their technical results in terms of the biological concepts of the system; this makes for very readable text, and I wish more in our field would do this.

We thank the reviewer for these kind comments.

I have two major comments, both of which are fairly philosophical, and then a small handful of minor comments.

The first major comment is that this analysis is conducted at the family level, for families with 10 or more species. I am worried that this may be insufficient statistical power to detect relevant trends, particularly in a phylogenetic framework. (I say this hypocritically, as I too have published on family-level analyses.) I would strongly encourage the authors to more explicitly explain why they think that “family” is the correct level to test their ideas at. I might also encourage (if it is not too much work), a set of analyses using a higher family size cut-off (20 species? even higher?), and/or to investigate what these patterns look like at the order level.

We appreciate the fact that the appropriate taxonomic level for testing Bergmann’s rule is controversial (see e.g., Salewski and Watt 2017). As the reviewer hints at, whatever choice one makes should provide ample statistical power to enable both meaningful comparisons and the detection of geographic patterns in body size. Higher taxonomic categories like Family or Order, group reasonably large numbers of species and hence enable more powerful analyses. However, as the phylogenetic distance between species within each group increases, we are more likely to see differences in morphology, natural history, etc., that can interact with body

size in unexpected ways and can preclude the detection of geographic size patterns like Allen's or Bergmann's rule (e.g., swallows and crows are very different types of birds yet both belong to the same order). This problem is particularly evident among Passeriformes, which include ca. 60% of all extant birds. We have therefore chosen taxonomic Family as the grouping level for our analyses because we believe it provides the best balance between the number of species per group and the diversity of morphologies, ecologies, behavior, etc. within them. We now add this justification to the main text in lines 56-60.

Nevertheless, we appreciate the concern that other readers may be curious about the potential effects of grouping levels in our findings so we also followed the reviewer's suggestions to increase our cutoff from 10 to 20 spp and to re-run all of our analyses using Order as our grouping level. The results for these two additional analyses can now be found in our Extended Data Figs. 6 h-j and Extended Data Table 2. As you will see, neither doubling the threshold nor using Order alters our main findings.

The second major concern is that a few of the authors' biological conclusions are not very clearly justified, potentially due to a word limit in the main text. Speculation on the many non-environmental pressures constraining beak size evolution, for example, or on why some clades adapt to thermoregulatory demands via Bergmann's Rule versus Allen's rule, is fascinating, and I think a real strength of this paper. More time explaining these assertions, however, would be prudent (especially lines 151-155).

Thank you for pointing this out. We were indeed concerned about word limits and tried to be as succinct as possible. We have added more detail as follows:

- Lines 123-126: On the logic behind possible non-environmental constraints on beak size

- Lines 127-136; 158-165: Better explanation on why some clades may adapt to thermal gradients through Bergmann's rule v. Allen's rule.

- Lines 166-178: We have reworded the last two paragraphs and the abstract, clarifying earlier statements and emphasizing our main conclusion that surface to volume ratio is the true variable of interest.

Minor comments:

By omitting the migratory species, there is going to be substantial geographic bias – do the authors think that their tilting focus towards tropical species, and away from temperate species, is affecting their results in any way? (To be clear, I find the decision to omit the migratory species to be well-justified, I'm just wondering what the biological repercussions of this decision are.)

We appreciate the opportunity to clarify this important issue. We excluded migratory species because migrants are likely to deviate from Bergmann's/Allen's rule and could therefore unduly inflate the number of "non-significant" patterns observed in our analyses. Specifically, migratory species leave their high-latitude breeding grounds during the coldest parts of the year, meaning that selection for bigger body sizes may not be as strong for them as it would be for resident species of similar breeding latitudes. To further complicate matters, migrants sometimes (but not always) exhibit very different wintering and breeding environments and, depending on the

timing of migration, different migratory species tend to differ in the degree to which they are exposed to cold during the highly variable “shoulder” seasons. Additionally, selection for larger body sizes in temperate habitats may be strongly constrained in migratory species because larger bodies can dramatically increase the energetic costs of flying.

For all these reasons, we consider that adaptation to temperature gradients in migratory birds deserves in-depth treatment in it of itself and have prepared a (second) companion paper to this submission that will properly and fully address these specifics. We are in the final stages of completing this companion paper and expect to be ready to submit within a couple of months.

In the meantime, we note that most families in our analyses include some year-round resident species in temperate and/or subtemperate habitats, allowing us to sample a reasonably diverse range of thermal environments in most groups (as both Allen and Bergmann had in mind when proposing their rules).

The figure quality in the reviewer copy is very poor, presumably something on the manuscript software’s end; something to be aware of for future submissions!

Thanks for alerting us about this issue! The resubmitted manuscript has high-resolution figures.

L119 – 25 is a strange citation for the multivariate selection occurring on beaks. Did you perhaps mean <https://doi.org/10.1098/rspb.2019.2474>, which has some overlap in authors? (Alternatively, if you want a citation on the competing effects of song and the environment on bird bill shape evolution, I quite like <https://doi.org/10.1111/ele.14102>) (And indeed, I don’t think you mean citation 25 in line 51 either?)

We thank both reviewers for alerting us about an issue with our reference manager software. We had meant citation 24 (which is the Friedman 2019 paper the reviewer suggests) and thank the reviewer for the extra Sebastianelli 2022 citation, which we have now added to the updated text (lines 50-51).

L307-301 – Using whose classification of migratory species? (Extracted from BirdLife’s ranges?)

We now provide these details in our methods (lines 334-338) and include our classifications with the raw data. Briefly, we classified a species as migratory whenever wintering or other part-time non-breeding ranges were reported in BirdLife (e.g., one wintering range and one year-round range). We note that this is a conservative definition of migration, as it eliminates both short- and long-distance migrants from our analyses and therefore minimizes biases against finding Bergmann’s/Allen’s rules.

L326: Indicate somewhere what EC, TND, BW, and BD mean (e.g., by putting the abbreviations in parentheses in lines 320-321).

This is now explained in lines 348-349 & 354 in the methods.

Data availability: Though no new data in the strictest sense were generated for this study, a substantial amount of work went in to generating morphological PC scores and

new environmental variables. I don't know the exact Nature Communications policy on data deposition, but I would expect these to be included for basic reproducibility and transparency.

Code availability: Including the code as a PDF is appreciated (it's more than many people do!), but it would be even better to deposit it in a format that can be easily copied and pasted. Furthermore, without the accompanying data (see above), it is not as helpful.

We appreciate this suggestion and have shared additional code (and associated data files) on Zenodo. As requested by the second reviewer, we have also included a table of family-level traits and conformance categories and family-traits (Data Table S1).

Code supplement: The use of red and green in the figures in the code supplement will make this very hard for readers with red-green colour-blindness to interpret; I strongly suggest selecting different colours.

Thank you for this suggestion. We now use a colorblind-friendly color palette following Crameri et al. 2020.

Crameri, F., G.E. Shephard, and P.J. Heron (2020), The misuse of colour in science communication, Nature Communications, 11, 5444. doi:10.1038/s41467-020-19160-7

Extended data figure 1: Are there photo credits for panel A? Or credits for the silhouettes in panels B-D?

Apologies for this omission. We now list all sources for photos and silhouettes (all under Creative Commons licenses) in our acknowledgements (lines 579-587) and have added the full list of sources to the archive on Zenodo as a table.

Extended data figure 2: What does the vertical blue line for the Rheidae mean?

Rheidae has only 2 species in AVONET which have similar tarsi lengths, so the resulting violin plot (which indicates density) appeared vertically "squeezed". In other words, what looked like a vertical line was actually a density plot (albeit, a not very informative one!). To alleviate this issue, we have replaced the vertical line with two circular points and clarified the caption (lines 647-652).

Reviewer #2 (Remarks to the Author):

NCOMMS-22-42817

Complementarity in Allen's and Bergmann's rules among birds

Broad comments:

In this paper, the authors undertake an interesting exploration of the combined influences of overall body size as well as the shape of organisms for thermal regulation. They seek to simultaneously estimate the support for two long-standing proposed macroecological gradients using existing morphological data from a large fraction of bird species from across the world. I appreciate the large-scale nature of this study and the efforts made to approach these questions from different angles using different morphological metrics, and their attempts to explain differences in responses among taxonomic groups. While I think there are interesting findings presented here, some quite crucial additional considerations are needed in multiple areas.

We thank the reviewer for their positive evaluation and their constructive feedback.

In particular, I had several major concerns pertaining to the bill length metrics used here and the implications these have for how results of this study are to be interpreted. Firstly, I have two points about relative bill size. There is a clear non-linear relationship between log bill size and log body mass (Fig. 2a). Modeling this using a linear regression therefore is not giving an accurate representation of relative bill size.

Thank you for pointing this out. We had initially followed standard procedures in the field, which typically involve using a linear model to account for the allometric scaling of the bill. However, you are entirely right in pointing out that the relationship in Figure 2a appears curvilinear so we followed your suggestion and recomputed bill residuals using a quadratic model instead. As expected, the curvilinear model of relative bill size indeed fits the data much better than the original linear model (see lines 79-80 and 378-387; delta AIC between models: 933; R-squared values improves from 0.486 to 0.5559 with non-linear fit, Extended Data Table 1). Accordingly, our manuscript now relies on relative bill sizes derived from a quadratic model in all downstream analyses and includes a revised Figure 2b as well as Extended Data Figures 5 and 9. All of our findings remain qualitatively identical (in fact, support for the patterns we had reported earlier is somewhat stronger).

Additionally, the authors make the case for the importance of bill surface area, so why not use that directly (estimated directly from bill measurements available in AVONET) rather than use length?

Measuring Surface to Volume ratio is difficult and hence most scholars have attempted to estimate it instead. However, as we now explicitly mention in the main text, given the diversity of bill anatomies, estimating the surface area of the bill from linear measurements is notoriously inaccurate. Similar issues plague the estimation of whole-body surface area, so what appears to be most common at the present (see Perez et al. 2014. Biol Open 3 (6): 486–488), is to estimate that quantity from body mass anyway (which, if anything, would introduce an additional source of error into our analyses). In conclusion, we still see great value in using the traditional proxies (bill length and body mass) until better metrics become available, but we caution in our

paper that these proxies need to be understood as such and must be properly contextualized when interpreting the results.

The authors ultimately conclude that relative bill length isn't the best metric to be used here, since any change in relative bill size might be due simply to changes in body size. I found it somewhat strange that such a large portion of the paper is dedicated to talking about relative bill size (throughout the manuscript) when this is ultimately rejected as being a valid approach by the authors (their thoughts, though not necessarily mine). The authors decide to use absolute bill length as an alternative metric. However, there are other issues with the use of this metric. Absolute bill size is likely to vary with body size for reasons unrelated to temperature (e.g., developmental or dietary reasons).

We appreciate the reviewer's concern that some aspects of bill size variation could be related to developmental mode and diet and have made our best effort to account for those effects. In terms of developmental mode, a literature survey indicates that we can expect that precocial birds grow their bills faster than altricial ones because they need to be able to feed themselves soon after hatching and larger bills may facilitate handling a wider variety of prey items. However, we could not find any prediction that relates developmental mode with the ultimate (i.e., adult) bill size. Considering that our analyses are based exclusively on adult specimens, we did not consider this factor further. As for diet, we have included a 5-category diet variable from Avonet in our models of bill size (lines 75-81 & 375-387). Once more, our main findings remain unchanged.

The fact that there is a negative relationship between bill size and temperature (i.e., opposing Allen's Rule) for bird families that show a strong negative relationship between temperature and body size (i.e., supporting Bergmann's Rule) supports this notion. That is, for families where species are larger in colder environments, they also tend to have larger bills in these environments (negative relationship between these two slope estimates as presented by L 93 and Fig. 2c). As traits tend to scale with body size, this isn't surprising. The question is, do these bill sizes vary more than would be expected under some null expectation of how bill size varies with body size? Without assessing deviations from this null expectation (what residuals are typically trying to get at), I don't think an accurate assessment of Allen's Rule can be made here. Given that this 'trade-off' between Bergmann's and Allen's Rules are a main focus of this paper, I think this stands as a major issue in this manuscript in its current form. I'm not sure there is a panacea to the fact that changes in relative bill size could be due to changes in body size alone.

We thank the reviewer for outlining these concerns so clearly. It has given us the opportunity to clarify our message and improve the manuscript. We understand and appreciate that relative metrics are critical to tease apart how traits that scale allometrically with body size may influence other traits or processes. We also understand that it is precisely for this reason that many earlier tests of Allen's rule used relative rather than absolute bill size. Nevertheless, the point we wish to make in our manuscript is that body size and bill size are *not the actual traits driving* the patterns that Bergmann and Allen observed... they are simply convenient proxies for surface-to-volume ratio (hereafter, SVR). Both authors were explicit about this in their original formulations of their rules: the ultimate reason for the patterns they observed was likely that

these traits relate to SVR and SVR enables increased heat loss/retention in warmer/colder habitats.

So why not just measure SVR and deal with it directly? We partially answered this question earlier. To begin with, this quantity is difficult to measure (even with modern tools) and estimating it from basic geometric shapes (i.e., assuming that spheres describe bodies and cones describe bills) is often very inaccurate and unequally biased across clades with different bill/body morphologies. The value of proxies like body and extremity size is that despite being imperfect estimates of the true quantity of interest, they are conspicuous, easy to measure, and correlated with SVR. We believe that this subtle, but important point has been lost over the years and that many recent papers overlook the fact that these metrics are proxies and treat them instead as if they were the true quantities of interest.

Our paper therefore begins by reminding readers that bills are likely to make contributions to heat dissipation/conservation that are not properly captured by body mass proxies because they are hollow (i.e., they weigh much less than other body parts of similar size) and are extensively vascularized. Based on this realization we pivot our analyses to ask then how can we achieve a better proxy for the SVR of the bill, through relative or absolute bill size? The answer is clear: absolute bill size is correlated with SVR whereas relative bill size is not. In other words, we can estimate the missing component of SVR through absolute and we can't do that through relative bill size which is what we do in the main text. We have incorporated some of these explanations in the main text to clarify our thinking (e.g., see lines 91-97).

One more important point of clarification. The reviewer mentions that families that show significant support for Bergman's rule show significantly larger bills in colder environments. Please note that this only applies 2.8% of all families in our sample (i.e., just 3/107 families show this pattern; see Table 1, right hand column). Given the extremely low frequency of this association, we hesitate to make a strong generalization as suggested.

It does appear that species that have positive support for Allen's Rule (larger bills in warmer areas) tend to have positive support for Bergmann's Rule (smaller bodies in warmer areas) – the caption states that no correlation was found, but I was unclear what analysis was used to conduct this analysis (see also detailed comments below).

We thank the reviewer for the opportunity to clarify our results here. We have now included more details on the metaregression in the caption for Fig. 2b to show that there is indeed no significant correlation between slopes when using relative bill size.

Body size may simply be less evolutionarily constrained than bill size, due to things like dietary factors (bill size and shape have been shown to be important for foraging, e.g., Galápagos finches, hummingbirds) and bird song (due to the importance of sexual selection). I think something that could be presented is how relative bill size change is related to absolute bill size change to assess this (that is, a scatterplot with these two metrics on the two axes).

We appreciate the reviewer's concern that body size could experience fewer evolutionary constraints than bill size and have explicitly tested this idea:

Specifically, the above figure depicts phylogenetic signal and ancestral state reconstructions of body mass, absolute bill size and relative bill size. Please note that phylogenetic signal was estimated via Blomberg's K and that p-values were obtained from 1,000 bootstrap replicates in phytools. Our analyses indicate that opposite to the reviewer's intuition, phylogenetic signal in body size (Blomberg's K = 4.094) is ca. four times higher than either that of absolute (Blomberg's K = 0.960) or relative bill size (Blomberg's K = 0.912). We also note that the strength of Allen's rule based on absolute bill size is significantly, yet only moderately correlated with the strength of Allen's rule based on relative bill size (Rho from Pearson's correlation coefficient test = 0.550, P < 0.001).

These two findings, however, seem to be only tangentially relevant to our manuscript and the questions we address. Considering that we are already covering a lot of material in our manuscript as is, we have elected not to include them in the revised version. We are happy to put them in if you consider it critical.

Additionally, the authors assume here that species are less impacted by shifting their morphology according to Allen's and Bergmann's Rules simultaneously (L 48, 106), compared to just one of these traits. I agree that changes in these traits could have important ecological implications (for several reasons). But if an organism is changing in two traits, then it is being impacted in two different ways. Just not as prominently in one single way (if there is some trade-off there, as we might expect). Is there really any buffering going on here by changing in two ways rather than one? That is, is it better to be slightly hampered in two different ways than more substantially hampered in one way? Possibly it is, but I don't think any evidence of this notion is presented here (nor can I think of a good way to actually explore the degree to which this might be true).

We concede that it is impossible to determine with certainty at this point whether subtle changes in two traits are less consequential to a lineage's expected fitness than a major change in a single trait. We nevertheless note that we are not assuming this is true. Rather we are simply *proposing* that this could be the case (note that minor variation in both body and bill sizes are already observed within populations and that those relatively small differences are not typically prohibitive). New ideas must start somewhere and while it is true that we cannot cite prior studies that already suggested what we propose, we have nevertheless demonstrated that the data on birds at least match our expectation.

I think changes in both size and appendage size are probably due to the fact that evolution acts on multiple axes at once. Maybe some species are more constrained in one area compared to another (possibly due to constraints due to diets, etc. as acknowledged in several places in the manuscript, including L 118). I would suggest rethinking how this is presented.

We certainly agree with the notion that evolution is acting on multiple axes... that is in fact an important point we were trying to get across with this study (e.g., see the final paragraph of our main text). We also agree with the notion that a lineage's characteristics, context, etc., may ultimately determine whether and how strongly it tends to adapt to thermal gradients via body size changes, bill size changes, or both. On that regard, please note that we explicitly attempted to figure out exactly which constraints/reasons/traits/etc. could be affecting these patterns through phylogenetic regression and randomization tests at the end of our manuscript. We find it encouraging that the reviewer has arrived at similar conclusions after reading our manuscript. Nevertheless, this comment suggests that we had not been sufficiently clear in the presentation of these ideas so we have reworded relevant sections of the main text to lay out these ideas more explicitly (see lines 116-136).

I do appreciate the consideration of how species' responses vary according to different traits and family characteristics. The idea that unusual bill sizes and shapes might be related to these patterns is interesting. The fact that families in warmer areas tend to correspond more with Bergmann's Rule is interesting. This suggests that it's these very warm areas that matter most for body size. This is in line with some recent work done

within-species, where larger effects of temperature were found in warmer parts of species' ranges (Youngflesh et al. 2022) and on the consequences of warming for body size in desert environments (Riddell et al. 2019).

Thank you for the kind comments. We must point out, though, that the variable we used in these analyses was not the mean temperature at which the species in a family tend to live. Rather, we use the temperature *range* of a family, meaning that larger values shall not be interpreted as indicative that species in a family live in warmer habitats but rather that the family members (as a whole) occupy a wider variety of thermal niches. The connections suggested in this comment therefore do not apply.

I do, however, think the analyses presented in Fig. 2d/e/f/g/h/u should be analyzed using continuous values. That is, the actually slope estimate (with associated uncertainty) for the Rule support used as the response variable, while temperature/# of species/bill shape used as the response variable. While presented in terms of a binary classification, support for these rules is actually a continuous measure.

Our original intent with these analyses was to determine the conditions, factors, or traits that could explain why practitioners sometimes reach the conclusion that some families follow Bergmann's rule, others that they follow Allen's rule, and most others, that they do not follow either rule. However, your comment made us realize that to properly answer this question it is imperative to also know what determines how strongly body or bill sizes change over thermal gradients (see added text in lines 127-136, 144-149). Thank you for this insightful suggestion.

As suggested, we have now added new analyses that use slope estimates as our response variable (see lines 144-149, 519-529). The conclusions that can be drawn from the combined results of the analysis that explore slope strengths with those that explore statistical "significance" are quite informative. Specifically, we find that the number of species in a family is not a significant predictor of effect size, clarifying that the reason why we tend to find significant Bergmann's rule in large families it is not because these families are more strongly affected by temperature than small families but rather because it is easier to detect similarly small effects when we have more statistical power! Similarly, we find that mean temperature range is a significant predictor of overall effect size, which indicates that the more extreme temperature differences that species within a group experience, the easier it is to detect changes in their morphology. We also find that bigger bills and bigger bodies are respectively related to stronger Allen's and Bergmann's gradients, presumably because when size is large to begin with, more pronounced changes in morphology are needed to achieve similar percentwise differences in SVR in comparison to small-bodied lineages (and these stronger effects are easier to detect). We added these results to the supplement and referenced them in our main text.

NB: we assumed here that the reviewer's characterization of temperature range/# of species/bill shape as "response" variables in this comment was a typo (i.e., that they meant "predictor")

I think more clarity regarding the statistical models is needed as well (both the overall relationship between temperature and both body size and bill size as well as in assessing the relationship between support for Bergmann's Rule and Allen's Rule). That is the authors could explicitly write out the full statistical model used in mathematical notation. For example, $y_i \sim N(\alpha + \beta x_i, \sigma)$ would be used to denote a simple linear model, where i is the index for observation. I bring this up

because the structure of the model has interpretations for hoe results are interpreted. I assume that a separate slope and intercept are estimated for each family in the initial analyses, but it's unclear in the text.

We thank the reviewer for this suggestion and have added mathematical notation to our different modelling approaches in the supplement (lines and 368-388 and 475-492).

Overall, while I enjoyed aspects of this paper and appreciate many of the ideas presented, I think additional considerations are needed. In particular, in my opinion the authors should rethink how these questions are approached and how results might be interpreted because of this. I think additional polish of these ideas and of the presentation are necessary before this could be acceptable for publication.

We appreciate the opportunity to improve the robustness of our analyses and the clarity of our presentation. We believe that the reviewer's comments have allowed us to better understand where we were falling short and hope that the changes we implemented to our analytical methods and text have properly addressed their concerns.

Minor comments:

General: I noticed the citations numbers are off in the manuscript. For example, citation 36 is not about brain size at L 60 – I believe that should be citation 35.

Thank you for catching this error. We have double-checked all citation numbers in the new manuscript and corrected earlier inadvertent mistakes.

General: I appreciate that the authors published the environment (session info with OS and package versions) used in the analyses. Information on the specific environment is not included enough, in my opinion. As a very minor point, I don't think all trace plots or Rhat values from the Bayesian model are needed in the supplement. I think including them distracts from the rest of the code.

We thank the reviewer for the positive comments and for the helpful suggestions. In response to these comments and those of Reviewer 1, we have abbreviated the lengthy sections detailing Rhat values and traceplots in the new supplement by providing stand-alone R scripts for each element of the results as requested by Reviewer 1 to facilitate portability. Information on the output of `>sessionInfo()` is now copy-pasted at the end of each R script for results shown in the main text.

Extended Data 1: Why have dashed lines only for ducks? Shouldn't there also be dashed lines for bill size in owls and body size for flycatchers?

We thank the reviewer for their careful observation. We have now added loosely and densely dashed lines to the figure and explained the interpretation of line types and colors in the caption of Extended Data Figure 1.

Fig. 1: It's a bit difficult to tell that the size of the 'birds' is different as well in panel C. I might suggest making the difference more pronounced.

Thanks! The differences are meant to visualize that (a) differences in body size can be subtle (as compared to differences in bill size) and (b) when both bill and body are changing together both differences can be even more subtle. We have clarified that the differences in bird cartoons are deliberately subtle in the caption for Figure 1 (line 284-286).

L 53: suggest removing 'comprehensive'. This is a large-scale study with a lot of value, but it is not a comprehensive analysis of morphology.

We understand the confusion that this statement generated and realize now that our wording was unclear. Thank you for pointing this out. Our study is indeed not a comprehensive review of morphology, but in terms of phylogenetic, latitudinal, and taxonomic coverage, it leaves only very few species unsampled (as reviewer 1 pointed out). The adjective comprehensive was meant to be applied to the latter, and we have now rephrased this sentence to make that connection more explicit (line 53).

L 56: I would suggest specifically stating that you looked at variation across species, within families, rather than simply saying 'family-wide' to make it more clear what was done.

We have incorporated this suggestion by rewording the sentence as follows (line 61-63).

"We begin by running two phylogenetic linear mixed models³⁵ to independently estimate interspecific variation in body and bill size within families, across geographic gradients of mean annual temperature"

ED Fig. 4: I'm not sure the information presented in panel d is very useful. We wouldn't expect relative bill size to correspond to bill surface area, but to relative bill surface area. That might be a better metric to have on the y-axis.

As noted above, the point of these plots is to show that absolute bill size is a better estimate of the component of an individual's heat transfer potential that is not properly captured by measures of body mass alone. In conversations with a variety of colleagues we have noticed that not everyone is aware of the fact that relative bill size is not correlated with surface area, so we stand by the inclusion of this graph (i.e., we believe it serves an important didactic purpose).

ED Fig. 5: If this is using simulated data, this should be stated in the caption. Is there an assumption being made in the generation of these data that bill size varies with body size linearly? Is that a good assumption? Would we expect that? I'm not sure I would as bill length is a one-dimensional measure and mass is often considered to be proportional to volume, which is a three-dimensional measure (and these things tend to scale with their dimensionality).

ED Fig. 5 plots actual data from specimens (as currently stated in the figure legend). Perhaps the reviewer meant Extended Data Figure 4 instead? ED Fig 4 was indeed created with simulated data. We have clarified this in the text and have included more details in the figure's caption (lines 665-669).

L 426: I think it's important to write out this model explicitly, using mathematical notation. I assume that the authors model 'observation error' in both the response and predictor here (with the response being the posterior mean values and the 'error' represented as the posterior standard deviation [or variance if that is used for the subsequent model]), but this is somewhat unclear. Were all data points used here or just those deemed to 'have an effect'?

We thank the reviewer for this suggestion and now provide more details on our modelling approach in lines 475-489. By accounting for uncertainty in both the response and the predictor we were indeed able to use all available data points, not just the ones that were previously deemed significant. We have now clarified this in lines 101-104.

ED Table 1: What do these numbers represent? The number of families?

Apologies for this omission. We now explain in the table's caption that these numbers refer to the number of families that conformed to each category (line 608).

General: It might be nice to have a table in the supplement that describes how many species each family had in the filtered dataset, and perhaps the general characteristics of that family (thermal range, total area occupied by that family, etc.)

Thanks for the suggestion. We now provide a table that contains summary information for each family (Supplemental Data Table 1 in data archive on Zenodo). Although Fröhlich et al. 2023 used total area occupied per species as a proxy variable for their study, we do not, as we consider the breadth of the temperature range (which is provided) a more pertinent variable, see point above.

Fig. 3: I think the bill characteristic images on the x-axis are somewhat confusing, in that they are distributed across the three panels. Additionally, I'm not sure it's appropriate to display each species as a point here. Conformity to these Rules are characterized at the family, not species level. This is a sort of visual pseudo-replication issue, in my opinion. Best to take the family average for these bill traits (which might also make the plots more clear).

We appreciate this comment and can see how our previous graphics were confusing. To address this issue, we have replotted figure 3a-c using family-level averages (as suggested), have modified the way in which extreme bill shapes for each PC are displayed and have added some text to the figure caption (e.g. line 315).

Reviewers' Comments:

Reviewer #1:

Remarks to the Author:

First of all, I wish to praise the authors for their thoughtful, thorough engagement with the reviewers' comments – this was skillfully done. This was already a very impressive and convincing study, and after this round of revisions the manuscript is even stronger.

My original concerns were relatively minor, and the authors have entirely addressed them. Reviewer 2 had far more to say; I'm sure this reviewer can speak for themselves, but for what it's worth, I have read through the authors' response to Reviewer 2 thoroughly and have nothing to add.

I have but two incredibly minor suggestions remaining:

Table 1 – it took me a very long time to figure out how to interpret this table, especially the parenthetical. Perhaps this is unavoidable (and/or my fault), but might a rephrase of L265-266 along the lines of "Of the slopes found non-significant for both criteria, the count of families with slope values nevertheless in the expected direction for both biological rules is shown in parentheses" might aid reader interpretation.

Table 2 – not all values are consistently rounded (Allen temperature range expected mean, Bergmann ln N per species observed mean, Neither ln N per species expected mean)

Congratulations to the authors for putting together a manuscript simultaneously so detailed and so elegant!

Reviewer #2:

Remarks to the Author:

NCOMMS-22-42817A

Complementarity in Allen's and Bergmann's rules among birds

Broad comments:

This is a resubmission. I also reviewed the initial version of this manuscript. Overall, I feel the authors have done a thorough job at considering and responding to my comments (as well as the comments of the other reviewer). The additional analyses and clarifying points in the response and the manuscript have satisfied my concerns with this work.

My only remaining comment is about the model fit for log bill size as a function of log body mass. It looks as if the model fit for Fig. 2a seems to be slightly 'off', but that could be due to the phylogenetic structure in the data that is accounted for in the model. It appears that the authors used a quadratic to model this association (unless I'm mistaken), which may not be the best choice given that this curve must 'come down' at some point (which we would not expect here). This, however, is a minor point and not really of much consequence to the major results or conclusions of this work. I appreciate the authors' reconsideration of their initial choice of a simple linear model here.

REVIEWERS' COMMENTS

Reviewer #1 (Remarks to the Author):

First of all, I wish to praise the authors for their thoughtful, thorough engagement with the reviewers' comments – this was skillfully done. This was already a very impressive and convincing study, and after this round of revisions the manuscript is even stronger.

My original concerns were relatively minor, and the authors have entirely addressed them. Reviewer 2 had far more to say; I'm sure this reviewer can speak for themselves, but for what it's worth, I have read through the authors' response to Reviewer 2 thoroughly and have nothing to add.

I have but two incredibly minor suggestions remaining:

Table 1 – it took me a very long time to figure out how to interpret this table, especially the parenthetical. Perhaps this is unavoidable (and/or my fault), but might a rephrase of L265-266 along the lines of “Of the slopes found non-significant for both criteria, the count of families with slope values nevertheless in the expected direction for both biological rules is shown in parentheses” might aid reader interpretation.

Table 2 – not all values are consistently rounded (Allen temperature range expected mean, Bergmann In N per species observed mean, Neither In N per species expected mean)

Congratulations to the authors for putting together a manuscript simultaneously so detailed and so elegant!

Author response

We thank reviewer 1 for the very kind appraisal and constructive suggestions. For the comment on the legend of Table 1, we have heeded the advice and added the suggested line in the caption for table 1 (lines 561-564). We also have updated the digits in Table 2 as suggested, so that all numbers have three digits after the decimal point. Thanks for catching this, we think it was produced by trailing zeroes being automatically dropped in our spreadsheet formatting. Now the format should be consistent.

Reviewer #2 (Remarks to the Author):

NCOMMS-22-42817A

Complementarity in Allen's and Bergmann's rules among birds

Broad comments:

This is a resubmission. I also reviewed the initial version of this manuscript. Overall, I feel the authors have done a thorough job at considering and responding to my comments (as well as the comments of the other reviewer). The additional analyses and clarifying points in the response and the manuscript have satisfied my concerns with this work.

My only remaining comment is about the model fit for log bill size as a function of log body mass. It looks as if the model fit for Fig. 2a seems to be slightly 'off', but that could be due to the phylogenetic structure in the data that is accounted for in the model. It appears that the authors used a quadratic to model this association (unless I'm mistaken), which may not be the best choice given that this curve must 'come down' at some point (which we would not expect here). This, however, is a minor point and not really of much consequence to the major results or conclusions of this work. I appreciate the authors' reconsideration of their initial choice of a simple linear model here.

Author response

We appreciate reviewer 2's positive assessment of our resubmission and thank them for their careful comments that have allowed us to improve our paper.

We agree that additional nonlinear model parametrizations might improve metrics of model fit. However, given how much model fit improved (Supplemental Table 1), we were surprised that the downstream conclusions were not majorly affected (compare first and second submission). Therefore, we reason that exploring alternative non-linear parametrizations beyond the quadratic fit presented here might marginally improve model fit but are unlikely to alter the principal findings. Therefore, we welcome the idea of exploring additional model complexity, but think that this is ultimately beyond the scope of our paper. However, we too shared the concern about Figure 2a (now Fig. 3a) and thank the reviewer for bringing this to our attention. We agree with the reviewer that the apparent visual discrepancy between data density and model fit could be in part due to the phylogenetic structure of the data and model.